# Influence of Biochar Mixed into Peat Substrate on Lettuce Growth and Nutrient Supply

Álvaro F. García-Rodríguez [1,2], Francisco J. Moreno-Racero [1], José M. García de Castro Barragán [1], José M. Colmenero-Flores [2,3], Nicolas Greggio [4], Heike Knicker [1,5,*] and Miguel A. Rosales [2,3]

1   Department of Biogeochemistry, Plant and Microbial Ecology, Instituto de Recursos Naturales y Agrobiología de Sevilla (IRNAS-CSIC), Adva. Reina Mercedes 10, 41012 Seville, Spain
2   Group of Plant Ion and Water Regulation, Instituto de Recursos Naturales y Agrobiología de Sevilla (IRNAS-CSIC), 41012 Seville, Spain
3   Laboratory of Plant Molecular Ecophysiology, Instituto de Recursos Naturales y Agrobiología de Sevilla (IRNAS-CSIC), Adva. Reina Mercedes 10, 41012 Seville, Spain
4   BiGeA—Biological, Geological and Environmental Sciences Department at Interdepartmental Centre for Environmental Sciences Research (CIRSA), Alma Mater Studiorum, University of Bologna, Via S. Alberto 163, 48123 Ravenna, Italy
5   Solid-State NMR Service Unit, Instituto de Recursos Naturales y Agrobiología de Sevilla (IRNAS-CSIC), Adva. Reina Mercedes 10, 41012 Seville, Spain
*   Correspondence: knicker@irnase.csic.es

**Abstract:** The use of peat in traditional cultivation systems and in commercial nurseries is an environmental problem. In this work, we evaluated the partial replacement of peat with different amounts of biochar sourced from vineyard pruning as plant growing substrates. We studied its effect on the growth of lettuce plants under greenhouse and semi-hydroponic conditions. Substrate mixtures contained 30% (*v/v*) of vermiculite and 70% (*v/v*) of different biochar:peat treatments as follows: 0:70 (B0), 15:55 (B15), 30:40 (B30), 50:20 (B50), and 70:0 (B70). Higher biochar treatments increased the pH and electrical conductivity of the substrate, negatively affecting plant growth and germination (especially in B70). The partial substitution of peat by 30% biochar (B30) delayed seed germination but improved plant growth and nitrogen use efficiency (NUE), with shoots containing higher levels of organic nitrogen and nitrate. Moreover, it increased the water holding capacity (WHC) and led to an efficient use of nutrients. Our study demonstrates that biochar can successfully replace and reduce peat and N fertilizer consumption. This has the potential to promote more sustainable farming with positive impacts on both plant growth and the environment.

**Keywords:** hydroponic culture; *Lactuca sativa* L. var. Batavia; nitrogen use efficiency; nutrient balance; pot experiments; plant stress parameters

## 1. Introduction

The volume of organic substrate used by the nursery sector in the European Union is 34.6 million m$^3$ per annum, of which 27 million m$^3$ is peat. In Europe, especially in the Netherlands and Spain, soil-reduced and soil-less substrates represent the main intensive cultivation systems [1,2]. The major component of most traditional plant growing substrates is peat, a non-renewable resource. Peat is favored in horticulture due to its high porosity and water holding capacity (WHC) and other positive plant growth properties [3].

However, there are a number of serious environmental issues concerning the use of peat as peatlands are important ecosystems with respect to biodiversity and ecosystem function. They are major carbon (C) sinks but become a negative C source when excavated. Peat mining also involves bog drainage, which leads to increased bog biodegradation on its surface layer, resulting in CO$_2$ emissions, a major contributor to climate change and global warming [4,5]. In Europe, peat is primarily extracted in the northern temperate

regions. However, the primary vegetable production takes place in Southern Europe. The long-distance transportation of this peat further adds to its C footprint. There is therefore a multi-facetted global need for sustainable and low-cost plant growing substrate alternatives to peat [2].

In recent years, various peat substitutes have been developed. Among them, the most promising products are coir (coconut fibers), pine bark, composted green byproducts, coco peat, kenaf stems, poultry feathers, rice husks, cotton gin trash, or switchgrass ([6] and literature therein). Another proposed option is biochar [7]. Biochar is a C-rich material produced from biomass processed in pyrolytic conditions, i.e., oxygen-limited thermochemical process at temperatures between 350–1000 °C. According to the European Biochar Initiative, biochar is a heterogeneous but highly aromatic substance. It must show an atomic H-to-C ratio < 0.7 and the organic C content should be between 35 and 95% of dry mass [8].

Biochar was originally proposed as a soil amendment and a potential "C negative" solution to promote soil C sequestration. Its C sequestration potential is due to its high biochemical resistance against degradation [9]. According to the feedstock, biochar has a high porosity, providing large surface areas and high WHC. It can also be used to prevent ion loss from soil due to its cation and anion exchange capacity, improving ion-adsorption by surface functional groups [10]. These promising properties have stimulated efforts in recent years to assess the impact of biochar on the quality of both growing substrates and growth of crop and ornamental plants. The number of publications in this field have grown exponentially [11,12]. Over five peer reviewed articles have been recently published [10,13–16]. However, the use of biochar as a potential substrate amendment to improve plant growth has been widely tested but also highly discussed. Literature has shown both positive and negative results [16–18]. There is therefore a critical need to clarify the usefulness of this approach. Most studies reporting on biochar effects on plant growth are focused on the analysis of total biomass production [19–21]. Many of those studies indicate that some biochars have the potential to replace commonly used soilless substrate. However, the impact of biochar on plant growth depends not only on the percentage of biochar in the substrate, but also on its feedstock and physical/chemical properties [22]. Further, nutrients provided by the biochar, the presence of potentially phytotoxic compounds, and its impacts on pH and water retention can alter the growth condition in a biochar amended substrate [23].

It has been reported that biochar addition can have several effects on nutrient interactions in soils and growing substrates [24]. Most of the nitrogen (N) is taken up by plants through roots as nitrate ($NO_3^-$) and ammonium ($NH_4^+$). However, most wood derived biochars are characterized by low N content [25]. Phosphorus (P) and potassium (K) are also important nutrients that can be provided by the biochar. Adding biochar with mineral fertilizer application has been reported to increase plant nutrition in many species, in addition to increased N content and uptake efficiency [26]. Nevertheless, this may lead to levels that are excessive for plant nutrition during the growth stage and therefore requires attention [27]. Replacing peat with biochar not only reduces peat consumption and greenhouse gas (GHG) atmospheric emissions, but also contributes to the concept of circular economy by conserving resources and minimizing inputs and waste. Furthermore, it benefits nutrient recovery and use efficiency [17].

For a better understanding of how plant growth can be affected by biochar addition, there is a need to determine: (i) the mineral nutrient composition supplied according to the type of biochar used; (ii) the occurrence of potentially phytotoxic compounds; (iii) the impact on the substrate pH and WHC. These factors will ultimately define how biochar amendment will affect the plant's physiological and stress status [28].

To achieve this, we characterized the growth and physiological parameters of lettuce plants cultivated on substrate mixtures containing vermiculite (30%), and commercial peat that was increasingly substituted by biochar obtained from vineyard prunings in the range of 15–70%. We undertook a pot experiment lasting 31 days during which we monitored the germination rate and plant growth parameters. Those parameters included substrate

properties, such as pH, electrical conductivity (EC), WHC, and mineral nutrient content. We additionally determined plant stress parameters, plant nutrient content, and N forms as well as nitrogen use efficiency (NUE).

## 2. Materials and Methods

### 2.1. Physical and Chemical Analysis of the Substrates

The biochar feedstock was provided by Caviro-Enomondo (Faenza, Italy) and obtained from vineyard pruning, pyrolyzed at a temperature of 500 °C in an initial open air full combustion, and later the oxygen was limited up to finish. Total pyrolysis time was 4 h. The particle size of the final biochar was <5 cm. Sphagnum peat and vermiculite substrates were both purchased from commercial firms (Klasmann-Deilmann GmbH, Geeste, Germany). The pH ($H_2O$) of biochar, peat substrate, and substrate mixtures was measured in suspension with distilled water (1:5) with a Crison pH-meter Basic 20 (Barcelona, España), applying the method described by De la Rosa et al. [29] for carbonized material. Subsequently, the supernatant solution was separated by filtration (Whatman N° 2 filter) to measure the EC in the filtered solution using a Crison EC-meter Basic 30+. The bulk density (BD) of the materials was obtained by measuring the weight of 100 mL dry material. To determine the WHC, 6 g of each substrate sample was placed on a Whatman 2 filter in a funnel and saturated with distilled water. For 2 h, the water was allowed to percolate through the filter and the funnel. Then, the weight of the moist samples was measured. The weight difference between dry and moist sample was extrapolated for a duration of the experiment of 12 h according to De La Rosa et al. [29]. The percentage relative to the dry weight (DW) of the substrate sample resulted in the value for the maximum WHC.

### 2.2. Determination of Nutrients Content

The total C (TC) and total nitrogen (TN) were determined in triplicates using a A "Flash 2000 elemental micro-analyser" (Thermo Scientific, Bremen, Germany). Total mineral nutrients (B, Ca, Cu, Na, Fe, K, Mg, Mn, S, P, and Zn) were quantified for substrates and plant shoot tissues in duplicates from the extracts obtained after controlled acidic digestion with ultrapure nitric and hydrochloric acid of the samples in a DigiPREP Block Digestion Systems (SCP Science) using inductively coupled plasma-optical emission spectroscopy (ICP-OES) (Varian Inc., Palo Alto, CA, USA). The contents of $NO_3^-$ and $NH_4^+$ in the substrates and plant shoot tissues were yielded from 100 mg of dried sample mixed with 10 mL miliQ water after a treatment in an orbital shaker for 2 h at room temperature. After that time, the tubes were centrifuged at $3000 \times g$ rpm for 5 min. The ionic content was measured in the supernatant by colorimetric assays "Omega SPECTROstar" (BMG LABTECH GmbH, Germany). The $NO_3^-$ content of the extract was measured using the salicylic-sulfuric acid method [30] and $NH_4^+$ was determined with an adapted protocol from the colorimetric method described by Greweling and Peech [31]. Finally, the organic N content was determined from plant dry shoot tissues by the Kjeldahl method [32], from which the $NH_4^+$ content was subtracted.

### 2.3. Plant Cultivation and Experimental Design

Lettuce (*Lactuca sativa* L. var. Batavia) seeds were sown under greenhouse conditions at $24 \pm 2$ °C/$17 \pm 2$ °C (day/night) and at $60 \pm 10\%$ relative humidity (EL-1-USB data logger, Lascar Electronics Inc., Erie, PA, USA). A photoperiod of 16 h/8 h with a flux density of photosynthetic photons (Mean PAR) 300–350 $\mu$mol $m^{-2}$ $s^{-1}$ (quantum sensor, LI-6400; Li-COR, Lincoln, NE, USA) and a light emission of 9000–10,000 lux (Digital Lux Meter, LX1010B; Carson Electronics, Valemount, Canada) was applied [33]. Seed germination was carried out in pots (4 cm × 4 cm × 10 cm) containing five different substrate mixtures with the following biochar:peat:vermiculite proportions (v:v:v): B0 (0:70:30), B15 (15:55:30), B30 (30:40:30), B50 (50:20:30) and B70 (70:0:30). Vermiculite was used to improve water and nutrient availability for root development. For each mixture,

6 replicates were prepared. Before sowing, seeds were vernalized in a cold chamber at 4–7 °C for 3–4 days to synchronize their germination. Subsequently, 3 seeds per pot were sown (1 cm deep in the substrate) and pots were incubated in transparent closed boxes to preserve humidity. All pots of each treatment were placed into a plastic tray and watered with 200 mL per tray of a basal nutrient solution composed of 1.42 mM $KNO_3$, 0.625 mM $KH_2PO_4$, 0.053 mM $K_2HPO_4$, 2.29 mM $Ca(NO_3)_2$, 1 mM $MgSO_4$, 0.1 mM FeNa-EDTA, 0.1 mM $H_3BO_3$, 0.1 mM $MnSO_4$, 29 μM $ZnSO_4$, 0.11 μM $CoCl_2$, 0.1 μM $CuSO_4$, 1 μM $Na_2MoO_4$, and 5 μM KI. The solution was adjusted to pH 5.7 with KOH. All trays were put into a greenhouse and accumulated germinated number of seeds were counted and noted every day until all seeds were fully germinated as seedlings. Germinated seeds were considered as the first leaf was seen.

After 15 days after sowing (DAS), one germinated lettuce seedling of each pot was selected and kept in the substrate whereas all other seedlings were removed. The selected seedlings were similar with respect to health and size between the treatments and kept growing under greenhouse and semi-hydroponic conditions for 31 days. Plants were watered every 3–4 days following a semi-hydroponic regime (with pots placed on trays containing at least 1 cm height of solution). At the end of the growing experiment, each tray was watered with a total of 5 L of basal nutrient solution. Every day, trays were rotated clockwise to provide comparable light and environmental conditions for all plants during the experiment. Neither pesticides nor extra fertilizers were applied before or during these studies.

*2.4. Leaf SPAD Index and Quantum Yield*

SPAD and quantum yield (QY) measurements were conducted after 25 DAS using a portable SPAD 502 Plus Chlorophyll Meter (Spectrum Technologies, Inc., Plainfield, IL, USA) and a portable fluorometer (FluorPen FP-100; Photon System Instruments, Brno, Czech Republic), respectively. For each plant 3–5 photosynthetically active and fully expanded intermediate leaves were used for all measurements.

*2.5. Determination of Plant Biomass, Leaf Parameters and Nitrogen Use Efficiency*

Sampling was performed from each combination of biochar:peat treatment after 31 DAS. Different plant tissues were harvested separately, roots manually cleaned from the substrate, and weighted to obtain the fresh weight. All leaves of a plant per pot were laid on a white background and photographed to measure the total leaf area, by a pixel quantification with ImageJ2 Software with a high precision of 99.95–100% [34]. Data were obtained in $cm^2$. Subsequently, plant tissues were lyophilized to determine the dry weight (DW), and their water content (WC) was calculated by difference of the fresh-to-dry weight [35]. Specific leaf area (SLA) was obtained by dividing the total leaf area by the total leaf DW [36]. Succulence was calculated as the water content divided by leaf area [37]. Total N content represents the sum of N obtained by the Kjeldahl method (as described above) and the N that forms part of $NO_3^-$ molecules, expressed as mg $g^{-1}$ DW. Total N accumulation (TNA) was calculated as the result of total N content multiplied by total DW, and results were expressed as mg of N. NUE is commonly defined as the vegetative yield (DW) per unit of N available in the substrate, expressed as g DW $g^{-1}$ N [38].

*2.6. Statistical Analysis*

Statistical analysis was performed using the STATGRAPHICS Centurion XIX software (StatPoint Technologies, Warrenton, VA, USA). The Shapiro–Wilk (W) test was used to verify the normality of the datasets. Both "one-way" and "two-way" analyses of variance (ANOVA) were performed to determine significant differences between groups of samples. Multiple comparisons of means were determined by the Tukey's honestly significant difference (HSD) and multiple range test (MRT). Principal component analysis (PCA) was also performed.

## 3. Results

### 3.1. Properties of the Substrate

Compared to other biochars derived from wood residues, our biochar revealed a lower C content (Table 1), which according to Lehmann and Joseph [39] could be best explained with a high contribution of other minerals present in the feedstock.

**Table 1.** Elemental composition (C and N), C:N ratios, pH, electrical conductivity (EC), bulk density (BD), and water holding capacity (WHC) represented as weight per weight of dry sample of the used biochar and peat substrates (*n* = 3).

| Treatment | TC (mg g$^{-1}$) | TN (mg g$^{-1}$) | C:N | pH | EC (μS cm$^{-1}$) | BD (g cm$^{-3}$) | WHC (% w:w) |
|---|---|---|---|---|---|---|---|
| Biochar | 408.9 ± 0.5 | 0.5 ± 0.0 | 817.8 | 10.4 ± 0.0 | 1530 ± 17 | 0.3 ± 0.0 | 164 ± 24 |
| Peat | 510.0 ± 11.3 | 19.9 ± 0.2 | 25.6 | 5.7 ± 0.0 | 460 ± 5 | 0.4 ± 0.0 | 118 ± 29 |

As it is typical for biochars from woody feedstock, it has a low N content of 0.05% leading to a wide C:N ratio. The elevated pH and EC indicate a large contribution of salt. Bulk density (BD) and WHC of this biochar are in the upper range observed for biochars derived from woody residues [25]. The peat substrate revealed C and N contents of 51% and 1.99%, respectively, leading to a C:N ratio of 26. The pH, EC, and WHC values were considerably lower than those of the biochar, whereas its BD was relatively superior.

Peat substitution by biochar resulted in a slightly alkaline pH (Table 2). Values of 7.9, 7.6 and 7.7 were detected in the substrate of B0, B15, and B30 treatments, respectively, whereas B50 and B70 treatments denoted alkaline pH values of 8.1 and 9.3, respectively. Accordingly, biochar addition increased the EC from a minimum of 118 μS cm$^{-1}$ in B0 to 1129 μs cm$^{-1}$ in B70. Although biochar had a considerable WHC, the amendment of this material to the peat substrate had no significant impact on WHC of the potting substrate at 0 DAS. However, at 31 DAS, the B30 treatment showed higher values (Table 2).

**Table 2.** pH, electrical conductivity (EC and water holding capacity (WHC) of biochar:peat: vermiculite mixtures (v:v:v).

| Treatment | Biochar (v:v %) | Peat (v:v %) | Vermiculite (v:v %) | pH | EC (μS cm$^{-1}$) | WHC (% w:w) | |
|---|---|---|---|---|---|---|---|
| | | | | | | 0 DAS [a] | 31 DAS |
| B0 | 0 | 70 | 30 | 7.9 ± 0.1 bc | 117 ± 9 d | 211 ± 49 | 269 ± 4 ab |
| B15 | 15 | 55 | 30 | 7.6 ± 0.1 c | 215 ± 3 cd | 119 ± 16 | 272 ± 6 ab |
| B30 | 30 | 40 | 30 | 7.7 ± 0.0 bc | 371 ± 17 c | 107 ± 19 | 289 ± 6 a |
| B50 | 50 | 20 | 30 | 8.1 ± 0.1 b | 642 ± 29 b | 96 ± 10 | 242 ± 10 bc |
| B70 | 70 | 0 | 30 | 9.4 ± 0.1 a | 1129 ± 45 a | 108 ± 13 | 223 ± 6 c |
| *p* | | | | ** | * | ns [b] | *** |

[a] DAS: days after sowing; [b] ns: no significant differences: Values are means of *n* = 3. Values followed by different letters in the same column indicate significant differences according to Tukey's test. Levels of significance: "ns." *p* > 0.05. * *p* ≤ 0.05. ** *p* ≤ 0.01. *** *p* ≤ 0.001.

### 3.2. Plant Germination

Peat substitution by biochar delayed germination (Figure 1). After 15 days, 94% of the seeds sown on B15 and B30 germinated, which corresponds to the value obtained for B0. B50 and B70 showed a significantly lower final germination rate of only 61%.

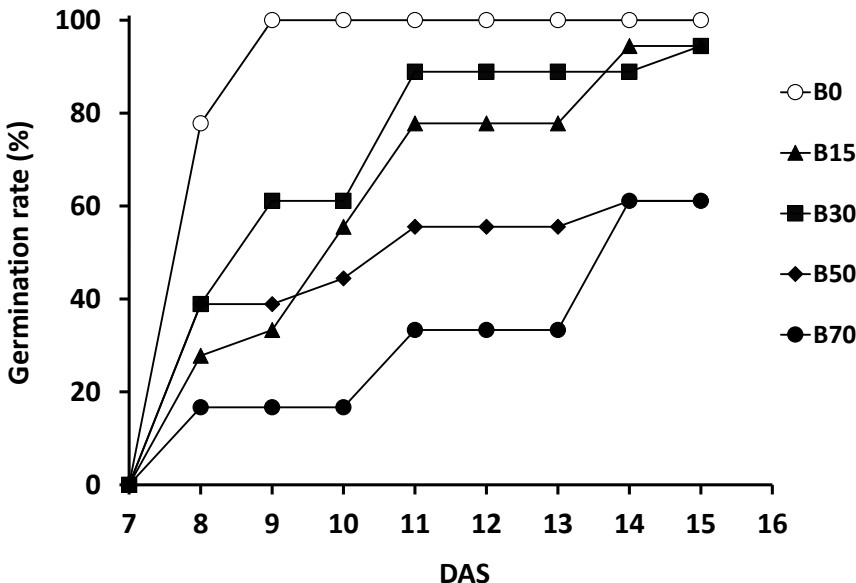

**Figure 1.** Germination rates of lettuce seeds sown on different biochar:peat substrate mixtures, with 0% (B0), 15% (B15), 30% (B30), 50% (B50) and 70% (B70) biochar contribution (*v*/*v*). Values are the accumulated percentage of seeds germinated at days counted. DAS, days after sowing.

### 3.3. Nutrient Dynamics in Substrates

At the beginning of the experiment (0 DAS), the total nutrient content of the different peat:biochar substrates was greatly affected by biochar addition (Tables 3 and 4). Thus, the contents of P, Ca, K, Cu, Zn, B, and Na, showed higher values in substrates containing biochar as compared with the peat substrate (B0). However, the contents of N and S decreased with increasing biochar contribution due to their low contents in the original biochar.

**Table 3.** Total content of macro-nutrients (N, P, S, Mg, Ca and K) detected in different peat:biochar substrates (Treatment: T) at the beginning (0 DAS) and the end of the experiment (31 DAS). Harvest Time (HT) is the time when substrate samples were collected at 0 and 31 DAS.

| | Treatment | N (mg g$^{-1}$ DW [a]) | P (mg g$^{-1}$ DW) | S (mg g$^{-1}$ DW) | Mg (mg g$^{-1}$ DW) | Ca (mg g$^{-1}$ DW) | K (mg g$^{-1}$ DW) |
|---|---|---|---|---|---|---|---|
| | Pure Biochar | 0.50 | 4.40 | 0.75 | 6.01 | 45.2 | 14.4 |
| **0 DAS [b]** | B0 | 13.62 ± 0.18 a | 0.45 ± 0.02 c | 1.68 ± 0.02 a | 32.42 ± 4.08 | 9.35 ± 0.28 c | 1.04 ± 0.07 c |
| | B15 | 10.73 ± 0.48 b | 1.39 ± 0.11 bc | 1.23 ± 0.20 ab | 35.60 ± 1.50 | 19.66 ± 5.58 bc | 3.11 ± 0.37 bc |
| | B30 | 7.70 ± 0.16 c | 1.29 ± 0.01 bc | 1.05 ± 0.12 ab | 32.68 ± 5.73 | 15.98 ± 0.32 bc | 4.81 ± 0.40 bc |
| | B50 | 4.49 ± 0.19 d | 2.34 ± 0.37 ab | 0.68 ± 0.11 b | 29.94 ± 3.24 | 28.09 ± 4.74 ab | 7.33 ± 0.73 ab |
| | B70 | 0.38 ± 0.02 e | 3.43 ± 0.32 a | 0.54 ± 0.04 b | 32.16 ± 2.35 | 41.88 ± 1.09 a | 11.11 ± 0.89 a |
| | *p* | *** | ** | ** | ns | * | *** |
| **31 DAS** | B0 | 13.92 ± 0.33 c | 0.64 ± 0.04 d | 2.36 ± 0.05 a | 44.43 ± 3.05 a | 12.50 ± 0.22 d | 1.90 ± 0.27 e |
| | B15 | 18.41 ± 1.60 bc | 1.41 ± 0.15 c | 1.98 ± 0.13 a | 48.27 ± 3.38 a | 19.45 ± 1.43 cd | 4.70 ± 0.34 d |
| | B30 | 24.19 ± 1.30 b | 1.78 ± 0.06 c | 2.14 ± 0.13 a | 35.84 ± 3.12 ab | 27.47 ± 1.59 c | 7.05 ± 0.18 c |
| | B50 | 36.26 ± 0.35 a | 2.76 ± 0.10 b | 1.36 ± 0.09 b | 36.93 ± 3.23 ab | 39.44 ± 2.36 b | 10.40 ± 0.27 b |
| | B70 | 36.64 ± 0.12 a | 3.85 ± 0.18 a | 0.65 ± 0.03 c | 31.47 ± 2.53 b | 57.34 ± 4.90 a | 12.20 ± 0.45 a |
| | *p* | ** | * | * | * | * | * |
| **T [c]** | | *** | *** | *** | ns | *** | *** |
| **HT [d]** | | *** | ** | *** | ** | *** | *** |
| **TxHT [e]** | | *** | ns | * | ns | ns | ns |

[a] DW: Dry weight; [b] DAS: days after sowing; [c] T: Level of significance with respect of treatment (biochar content); [d] HT: Level of significance with respect to harvest time (0 and 31 DAS). [e] TxHT: Level of significance with respect of the interaction of T and HT. Values are the mean of *n* = 6. Values followed by different letters in a column indicate significant differences according to Tukey's test. Levels of significance: *p* > 0.05 ("ns." not significant differences); * *p* ≤ 0.05. ** *p* ≤ 0.01. *** *p* ≤ 0.001.

**Table 4.** Total contents of micro-nutrients (Fe, Mn, Cu, Zn and B) and sodium (Na) detected in different peat:biochar substrates (Treatment: T) at the beginning (0 DAS) and the end of the experiment (31 DAS). Harvest Time (HT) is the time when substrate samples were collected at 0 and 31 DAS.

| | Treatment | Fe (mg g$^{-1}$ DW [a]) | Mn (µg g$^{-1}$ DW) | Cu (µg g$^{-1}$ DW) | Zn (µg g$^{-1}$ DW) | B (µg g$^{-1}$ DW) | Na (µg g$^{-1}$ DW) |
|---|---|---|---|---|---|---|---|
| | Pure Biochar | 2.42 | 250 | 147 | 169 | 47.9 | 735 |
| 0 DAS [b] | B0 | 16.15 ± 2.01 | 315.65 ± 43.21 | 74.74 ± 11.87 b | 48.45 ± 3.40 c | 1.12 ± 0.09 c | 145.95 ± 9.60 c |
| | B15 | 18.32 ± 1.54 | 352.11 ± 18.95 | 107.33 ± 14.70 b | 68.51 ± 4.60 bc | 9.18 ± 1.50 bc | 188.43 ± 19.26 bc |
| | B30 | 16.19 ± 3.74 | 301.34 ± 37.83 | 92.42 ± 13.32 b | 76.70 ± 10.06 abc | 12.58 ± 0.50 bc | 228.10 ± 9.10 bc |
| | B50 | 18.45 ± 5.86 | 472.61 ± 113.00 | 140.51 ± 25.32 ab | 173.39 ± 50.19 ab | 23.40 ± 4.11 ab | 370.67 ± 34.04 ab |
| | B70 | 15.35 ± 1.19 | 487.23 ± 43.72 | 196.73 ± 6.63 a | 179.85 ± 10.36 a | 37.58 ± 1.92 a | 486.23 ± 27.75 a |
| *p* | | ns | ns | * | * | *** | *** |
| 31 DAS | B0 | 22.31 ± 0.97 | 427.14 ± 18.19 b | 86.55 ± 6.11 c | 98.69 ± 3.60 d | 11.83 ± 0.81 d | 135.50 ± 14.80 d |
| | B15 | 22.30 ± 1.28 | 493.56 ± 21.88 ab | 127.29 ± 8.69 bc | 117.23 ± 5.30 cd | 20.54 ± 1.22 cd | 239.99 ± 21.99 c |
| | B30 | 19.70 ± 1.08 | 546.61 ± 19.26 ab | 142.44 ± 6.31 bc | 189.04 ± 24.06 bc | 31.98 ± 2.42 bc | 410.52 ± 18.62 b |
| | B50 | 21.46 ± 2.43 | 606.89 ± 57.49 a | 186.92 ± 12.92 b | 212.28 ± 12.16 ab | 43.93 ± 1.41 ab | 501.37 ± 25.97 ab |
| | B70 | 20.24 ± 1.11 | 629.08 ± 37.08 a | 253.07 ± 14.20 a | 283.02 ± 12.63 a | 51.41 ± 3.70 a | 524.52 ± 22.07 a |
| *p* | | ns | ** | *** | *** | *** | ** |
| T [c] | | ns | *** | *** | *** | *** | *** |
| HT [d] | | ** | *** | *** | *** | *** | *** |
| TxHT [e] | | ns | ns | ns | ns | ns | ** |

[a] DW: Dry weight; [b] DAS: days after sowing; [c] T: Level of significance with respect of treatment (biochar content); [d] HT: Level of significance with respect to harvest time (0 and 31 DAS). [e] TxHT: Level of significance with respect of the interaction of T and HT. Values are the mean of *n* = 6. Values followed by different letters in a column indicate significant differences according to Tukey's test. Levels of significance: *p* > 0.05 ("ns." not significant differences); * *p* ≤ 0.05. ** *p* ≤ 0.01. *** *p* ≤ 0.001.

The concentrations of Mg, Fe and Mn remained unaffected between treatments, possibly because comparable amounts were present in both feedstocks. At the end of the experiment (31 DAS), a comparable trend with respect to nutrient content and biochar supply was observed, although for all nutrients an accumulation is revealed with respect to 0 DAS (Tables 3 and 4). The latter is best explained with the addition of nutrients present in the fertigation solutions applied during the experiment. From this, it can be deduced that no nutrient limitation occurred in our experiment. Note that the behavior of the N contents in the substrates cannot be explained solely by the N addition with fertigation. The latter would enhance the N contents of all substrates by comparable amounts. The fact that increasing biochar concentration goes along with increasing N sequestration indicates that properties of the biochar are responsible for the accumulation of N in the substrate. The combined impact of biochar concentration (treatment: T) and experiment time (harvest time: HT) is confirmed by calculating the significance level of the interaction between T and HT (TxHT) in Table 3.

### 3.4. $NO_3^-$ and $NH_4^+$ Content in the Substrates

As expected, $NH_4^+$ and $NO_3^-$ contents of the biochar were below the detection limit (Table 5) which explains that at 0 DAS, the content of $NO_3^-$ decreased significantly from 0.26 mg g$^{-1}$ DW in B0, to 0.17 mg g$^{-1}$ DW in B70 with increasing biochar supply. A comparable and significant effect was deduced for $NH_4^+$ contents. Compared to the starting conditions, at 31 DAS, higher $NO_3^-$ contents were detected as the biochar rate increased in the treatments. The contents of $NH_4^+$, on the other hand, decreased five times from B0 to B70. When we delved into the statistical analysis of the $NO_3^-$ and $NH_4^+$ contents in the substrates mixtures, our results revealed that both $NO_3^-$ and $NH_4^+$ exhibited significant differences according to two factors: the specific substrates treatment at any harvest time (T) and the harvest time independently from substrate treatment (HT). Moreover, the interaction between these two factors was significant, indicating that in

addition to fertigation, biochar properties or biochar concentration must have had an impact on the inorganic N content in the substrate.

**Table 5.** $NO_3^-$ and $NH_4^+$ contents in biochar:peat mixtures at the beginning (0 DAS) and the end of the experiment (31 DAS).

| Treatment | $NO_3^-$ (mg g$^{-1}$ DW [a]) | | $NH_4^+$ (mg g$^{-1}$ DW) | |
|---|---|---|---|---|
| | 0 DAS [b] | 31 DAS | 0 DAS | 31 DAS |
| **Biochar** | Bd [c] | | bd | |
| **B0** | 0.26 ± 0.01 a | 0.55 ± 0.07 c | 0.24 ± 0.01 a | 0.05 ± 0.01 a |
| **B15** | 0.24 ± 0.01 ab | 0.67 ± 0.09 bc | 0.19 ± 0.01 b | 0.03 ± 0.01 ab |
| **B30** | 0.22 ± 0.01 b | 1.13 ± 0.10 ab | 0.14 ± 0.01 c | 0.02 ± 0.01 b |
| **B50** | 0.19 ± 0.01 c | 1.27 ± 0.11 a | 0.08 ± 0.01 d | 0.02 ± 0.01 b |
| **B70** | 0.17 ± 0.01 d | 1.42 ± 0.21 a | 0.01 ± 0.01 e | 0.01 ± 0.01 b |
| *p* | ** | * | ** | * |
| **T [d]** | *** | | *** | |
| **HT [e]** | *** | | *** | |
| **TxHT [f]** | *** | | *** | |

[a] DW: Dry weight; [b] DAS: days after sowing; [c] bd: below detection limit, [d] T: Level of significance with respect of treatment (biochar content); [e] HT: Level of significance with respect to harvest time (0 and 31 DAS); [f] TxHT: Level of significance with respect of the interaction of T and HT. Values are the mean of *n* = 6. Values followed by different letters in columns indicate significant differences according to Tukey's test. Levels of significance: * $p \leq 0.05$. ** $p \leq 0.01$. *** $p \leq 0.001$. DAS, days after sowing; DW, dry weight.

### 3.5. Plant Growth, Water and Stress Parameters

Among the five treatments, the greatest production of dry shoot and root biomass was recorded for B15 and B30 (Figure 2). Dry shoot biomass increased by 16.8% and 26.9% for B15 and B30, respectively, if compared to the substrate lacking biochar (Figure 2B). The dry root biomass responded with an increase of 27.9% and 42.4% (Figure 2C). When the biochar content reached 50% of the substrate volume (B50), both shoot and root plant biomass declined compared to those observed for B30 and reached comparable yields as B0. At 70% of biochar in the substrate (B70), plant biomass production was considerably reduced (Figure 2A). At the same time, this substrate led to the highest root to shoot ratio related to the dry biomass value of 2.05 (Figure 2D), indicating plant stress symptoms.

Table 6 shows that neither leaf water content nor succulence exhibited significant differences with respect to increasing biochar amendment. Note that plants grown under the B70 treatment are no longer included in the analysis of physiological parameters due to their extremely low growth (Figure 2). Compared to B0, leaf area was positively influenced by B15 and B30 treatments, increasing up to 18.8% and 30.3% relative to the substrate without biochar (B0). In contrast, the application of 50% biochar slightly reduced plant leaf area by 12.6% relative to B30, but it was still 20.3% higher than in the experiment without biochar addition. However, looking at the specific leaf area (expressed as cm$^2$ g$^{-1}$ DW), no significant differences among the different treatments were found. Regarding the plant stress indicators studied, SPAD measurements showed no significant differences between B0, B15, and B30 treatments, but B50 showed a significant lower SPAD index by 20.9% compared to the control treatment. However, quantum yield measurements did not show significant differences among treatments (Table 6).

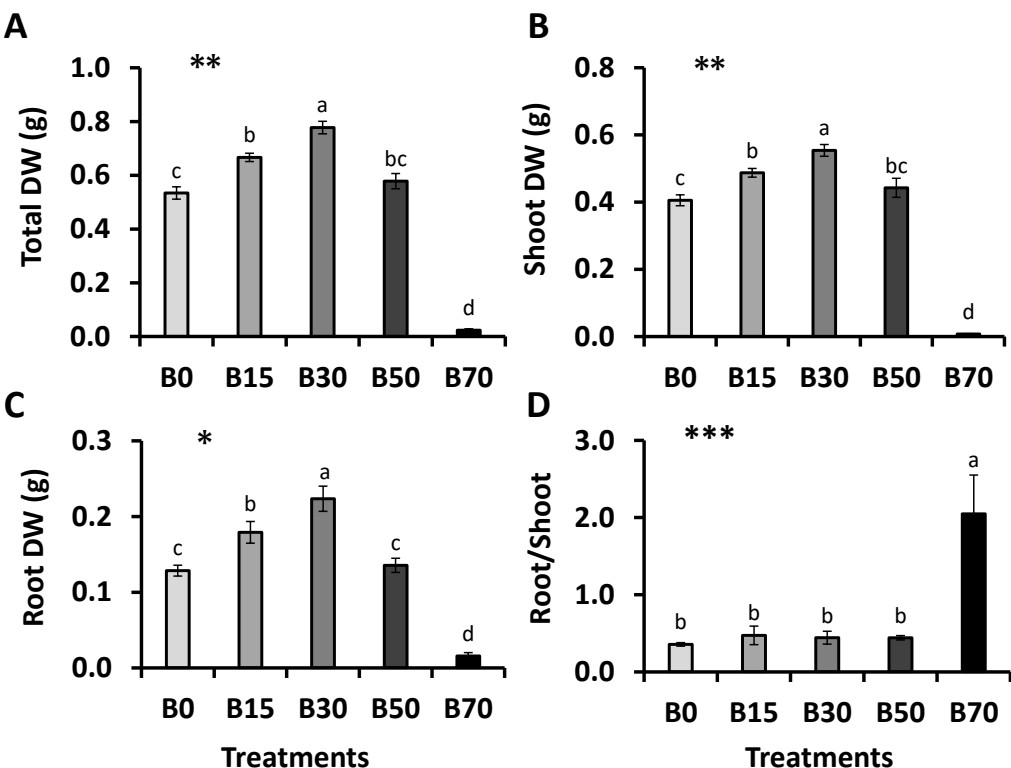

**Figure 2.** Growth parameters of lettuce plants cultivated under different peat:biochar substrates. (**A**) Total Dry Biomass (g); (**B**) Shoot Dry Biomass (g); (**C**) Root Dry Biomass (g); and (**D**) Root:shoot ratio (%). Values are the mean of $n = 6$. Values with different letters are significantly different according to Tukey's test. Levels of significance: * $p \leq 0.05$. ** $p \leq 0.01$. *** $p \leq 0.001$.

**Table 6.** Water and stress-indicating parameters in lettuce plants cultivated under different peat:biochar substrates. Plants grown on different biochar:peat mixtures.

| Treatment | Water Content (%) | Leaf Area (cm²) | Specific Leaf Area (cm² g⁻¹) | Succulence (mg H₂O cm⁻²) | SPAD Index | Quantum Yield (F'm F'v⁻¹) |
|---|---|---|---|---|---|---|
| **B0** | $95.48 \pm 0.13$ | $259.19 \pm 16.16$ b | $74.83 \pm 5.81$ | $31.68 \pm 1.72$ | $16.70 \pm 0.30$ a | $0.74 \pm 0.01$ |
| **B15** | $95.83 \pm 0.13$ | $319.98 \pm 12.72$ ab | $68.09 \pm 2.42$ | $35.31 \pm 1.84$ | $16.92 \pm 1.17$ a | $0.74 \pm 0.01$ |
| **B30** | $95.85 \pm 0.12$ | $371.80 \pm 30.40$ a | $63.14 \pm 5.31$ | $35.53 \pm 3.39$ | $16.87 \pm 0.58$ a | $0.75 \pm 0.01$ |
| **B50** | $95.63 \pm 0.27$ | $324.99 \pm 19.72$ ab | $67.82 \pm 2.01$ | $33.37 \pm 2.19$ | $13.17 \pm 0.59$ b | $0.74 \pm 0.01$ |
| *p* | ns | * | ns | ns | * | ns |

Values are the mean of $n = 6$. Values followed by different letters in columns indicate significant differences according to Tukey's test. Levels of significance: $p > 0.05$ ("ns." not significant differences); * $p \leq 0.05$.

### 3.6. Plant Nutritional Status

To better understand the effect of the partial substitution of peat by biochar on the yield of lettuce plants, the total content of macro- and micro-nutrients, as well as Na, was quantified. Different biochar treatments significantly affected the content of N, P, Mg, Ca, K, Fe, Mn, Cu, Zn, B, and Na in leaves of lettuce plants (Tables 7 and 8), whereas no impact on the S content was found. Interestingly, the content of N in the shoots increased with biochar treatments (Table 7), which is in line with the higher N content recorded in substrates containing biochar after 31 DAS (Table 3). Although the content of P in the substrate enhanced with biochar supply, no differences in the shoot P content were found between B0 and B30 plants, whereas it significantly decreased in those of the B50 treatment. Accordingly, the content of S, Mg and Mn did not change in either the substrate or the shoot of the B30 plants compared to those of B0. The B30 treatment induced a higher content of

K, Cu, and Zn in the shoots, which is consistent with the higher accumulation of K and Zn or no changes in Cu as it was recorded for the B30 substrate. However, the biochar supply (B30 and B50) significantly decreased the content of Ca, Fe, and B in the shoots, which was not correlated with their accumulation in the respective substrate mixtures. Overall, enhancing the biochar supply up to 50% augmented the content of Mg and Zn in the shoots but decreased the content of P, Ca, Fe, and B, and had no impact on those of S, K, Mn, and Cu, if compared with B0. Although Na is not considered an essential element for plants, its accumulation in plant tissues is commonly associated with toxicity under salinity conditions. In our experiments, the partial substitution of peat by biochar enhanced the Na content by 28% and 59% in the shoots of plants grown on B30 and B50 (Table 8), respectively. This is in line with 3- and 3.7-times higher Na content in B30 and B50 with respect to B0 which can be associated to the biochar addition (Table 4).

**Table 7.** Macronutrient (N, P, S, Mg, Ca and K) content detected in shoots of lettuce grown on different biochar:peat substrates.

| Treatment | N (mg g$^{-1}$ DW [a]) | P (mg g$^{-1}$ DW) | S (mg g$^{-1}$ DW) | Mg (mg g$^{-1}$ DW) | Ca (mg g$^{-1}$ DW) | K (mg g$^{-1}$ DW) |
|---|---|---|---|---|---|---|
| B0 | 41.51 ± 0.46 bc | 8.54 ± 0.30 a | 3.56 ± 0.16 | 4.59 ± 0.21 b | 12.95 ± 0.43 a | 71.09 ± 2.09 b |
| B15 | 39.39 ± 1.61 c | 8.72 ± 0.30 a | 3.90 ± 0.13 | 3.91 ± 0.12 b | 11.78 ± 0.51 a | 83.21 ± 2.47 a |
| B30 | 50.08 ± 2.00 ab | 8.59 ± 0.25 a | 4.38 ± 0.10 | 4.23 ± 0.11 b | 9.84 ± 0.37 b | 83.60 ± 1.50 a |
| B50 | 55.32 ± 2.88 a | 7.05 ± 0.47 b | 4.07 ± 0.36 | 5.67 ± 0.30 a | 7.87 ± 0.56 c | 76.65 ± 3.16 ab |
| *p* | * | ** | ns | *** | *** | ** |

[a] DW: dry weight. Values are the mean of *n* = 6. Values followed by different letters in columns indicate significant differences according to Tukey's test. Levels of significance: *p* > 0.05 ("ns." not significant differences); * $p \leq 0.05$. ** $p \leq 0.01$. *** $p \leq 0.001$.

**Table 8.** Micronutrient (Fe, Mn, B, Zn, Cu) and Na detected in shoots of lettuce grown on different biochar:peat substrates.

| Treatment | Fe (µg g$^{-1}$ DW [a]) | Mn (µg g$^{-1}$ DW) | Cu (µg g$^{-1}$ DW) | Zn (µg g$^{-1}$ DW) | B (µg g$^{-1}$ DW) | Na (mg g$^{-1}$ DW) |
|---|---|---|---|---|---|---|
| B0 | 266.85 ± 48.93 a | 156.90 ± 11.64 a | 6.77 ± 0.43 b | 64.43 ± 3.63 b | 36.11 ± 1.26 a | 1.16 ± 0.18 b |
| B15 | 160.24 ± 12.20 b | 89.06 ± 5.59 b | 10.69 ± 0.98 ab | 96.05 ± 8.07 ab | 28.57 ± 1.07 b | 1.15 ± 0.08 b |
| B30 | 115.97 ± 5.08 b | 140.39 ± 3.84 ab | 15.96 ± 4.46 a | 97.93 ± 5.86 a | 26.62 ± 1.36 b | 1.48 ± 0.11 ab |
| B50 | 132.10 ± 17.66 b | 190.81 ± 24.27 a | 13.02 ± 0.53 ab | 112.70 ± 12.01 a | 24.03 ± 1.58 b | 1.84 ± 0.09 a |
| *p* | ** | *** | * | ** | *** | ** |

[a] DW: dry weight. Values are the mean of *n* = 6. Values with different letters in columns indicate significant differences according to Tukey's test. Levels of significance: * $p \leq 0.05$. ** $p \leq 0.01$. *** $p \leq 0.001$.

### 3.7. Content of Different Nitrogen Forms and NUE

The shoot content of organic N, $NO_3^-$, and $NH_4^+$, as well as total N accumulation per plant (TNA) and NUE levels (dry weight of plant material per unit of available N in the substrate) manifested significant differences in relation to the amount of biochar amended to the different substrate mixtures (Table 9). Compared to plants grown on B0, those with the B30 treatment showed an increase of 14.1% and 65.8% in the content of organic N and $NO_3^-$, respectively, whereas the content of $NH_4^+$ did not show significant differences in any of the peat:biochar treatments (Table 9). Interestingly, the B30 treatment exhibited an increase of 28.8% and 29.4% in TNA and NUE levels, respectively. The content of $NO_3^-$ reached higher levels in the shoots with the B50 treatment, which is in line with $NO_3^-$ accumulation observed in the B50 substrate after 31 DAS.

**Table 9.** Contents of N forms ($NH_4^+$, $NO_3^-$, Organic N), total N accumulation (TNA) and nitrogen use efficiency (NUE); defined as dry weight (DW) per unit of available N in the substrate, in shoots of lettuce plants grown on biochar:peat substrates.

| Treatment | Organic N (mg g$^{-1}$ DW [a]) | NO$_3^-$ (mg g$^{-1}$ DW) | NH$_4^+$ (µg g$^{-1}$ DW) | TNA (mg N) | NUE (g DW g$^{-1}$ N) |
|---|---|---|---|---|---|
| **B0** | 33.32 ± 1.36 b | 7.62 ± 0.19 bc | 0.07 ± 0.01 | 30.32 ± 3.85 b | 0.91 ± 0.04 c |
| **B15** | 33.83 ± 0.70 b | 5.17 ± 0.42 c | 0.09 ± 0.01 | 29.21 ± 2.29 b | 1.46 ± 0.03 b |
| **B30** | 38.04 ± 0.59 a | 13.90 ± 2.09 ab | 0.09 ± 0.01 | 45.18 ± 2.13 a | 2.23 ± 0.12 a |
| **B50** | 37.17 ± 0.26 a | 18.01 ± 2.63 a | 0.10 ± 0.01 | 32.07 ± 0.78 ab | 1.53 ± 0.16 b |
| *p* | * | * | ns | * | ** |

Values are the mean of *n* = 6. Values with different letters are significantly different according to Tukey's test. Levels of significance: $p > 0.05$ ("ns." not significant differences); * $p \leq 0.05$. ** $p \leq 0.01$.

### 3.8. Principal Component Analysis (PCA)

Two principal component analyses (PCA) were performed in order to identify potential differences which measured parameters and peat:biochar treatments affected lettuce plant development (Figure 3A,B).

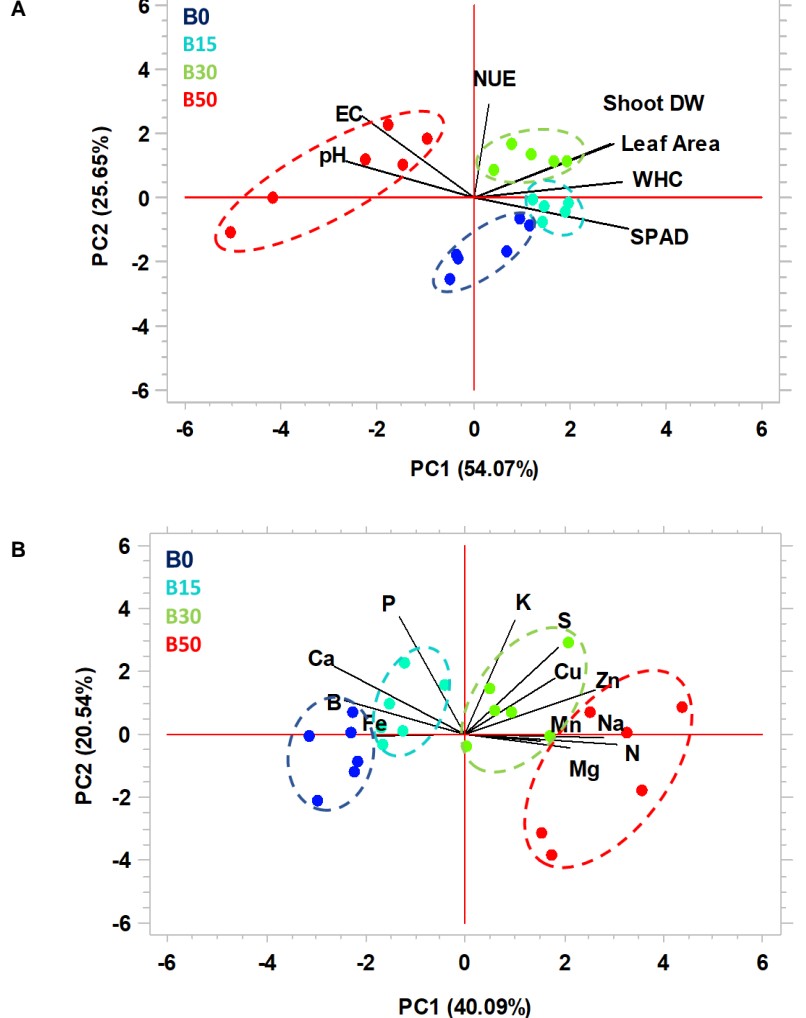

**Figure 3.** PCA biplot with the first two PCA axes, with projected centroids of peat:biochar treatments (B0-dark blue, B15-light blue, B30-green, B50-red). (**A**) Substrate parameters (pH, EC and WHC) and plant parameters (Shoot DW, Leaf area, SPAD, NUE) are shown. (**B**) Nutrients content (N, P, S, Mg, Ca, K, Fe, Mn, Cu, Zn, B and Na) detected in plants are shown.

The first PCA analysis was performed using both substrate (pH, EC, WHC) and plant (Shoot DW, Leaf area, SPAD, NUE) parameters. The first two axes explain 79.71% (PC1: 54.07%; PC2: 25.65%) of the total variance (Figure 3A). A second PCA was performed using the content of nutrients (N, P, S, Mg, Ca, K, Fe, Mn, Cu, Zn, B, Na), with the two axes explaining 61.45% (PC1: 40.09%; PC2: 20.54%) of the total variance (Figure 3B).

The effectiveness of the PCA results in interpreting the differences across different substrates and plant parameters in response to rising biochar exposure. In the current study, the loading matrix indicates that variation in substrate factors, such as pH and EC, are most closely aligned with the B50 treatment. In contrast to this, plants grown under the B30 treatment are positioned on the positive side of PC1, in the upper right quadrant of the PCA score plot, as they delivered higher shoot DW, leaf area and NUE. Moreover, lettuce plants were characterized by higher SPAD approaching to B15 treatment.

The score plot of the second PCA integrated useful information on the macro- and micro-nutrients, Na content, and peat:biochar treatments. Plants grown under B0 and B15 treatments were positioned on the negative side of PC1 in the upper and lower right quadrant of the PCA score plot as it delivered with high levels of B, Fe, Ca, and P, respectively. Plants grown under B30 and B50 treatments were positioned on the positive side of PC1 in the upper and lower right quadrants of the PCA score plot as this delivered high levels of K, S and Cu; N, Mg, Mn, and Zn, respectively.

## 4. Discussion

### 4.1. Substrate Characterization

The biochar addition to growing substrates has been shown to impact pH, EC, or ions of the media, and thereby to influence the availability of nutrients for plant uptake [40]. As stated by Wortman [41], the pH should be kept in the range of 5.5–6 to achieve optimal growth for most plants, although a higher pH of up to 8 may be tolerated. The high pH of 10 of our biochar may not be a problem if biochar addition is low enough to maintain the pH of the substrate mixture within a tolerable range for plants to grow. In our experiment, this is the case up to a content of 50% biochar. However, it decreased the overall performance of the plants and the beneficial effects of peat substitution by biochar (Figure 2). The presence of 70% biochar, however, induced serious problems for plant growth. The high pH level of our biochar is related to its high contributions of Ca and K, as it was also found in other wood-derived biochars presented in previous research [42]. Others [43] established reasonable relationships among biochar and soil properties, such as pH, cation exchange capacity, and exchangeable $Ca^{2+}$, $K^+$, and $Mg^{2+}$.

The EC is an indicator for the availability of nutrients. Some biochars contain high amounts of soluble salts. Hence, the EC of the biochar-amended substrate increases with the increasing biochar treatments. However, the EC of the substrate should be kept within the range which allows sufficient nutrient availability but does still not cause salinity effects [44]. In our experiment, similarly to the pH, the EC increased with the biochar content of the substrate mixture (Table 2), but major negative impacts on plant performance were particularly observed for the 70% biochar treatment relative to the only peat experiment.

Since biochar from wood is commonly composed of particles with a size in the range of centimeters, its application is likely to alter properties, such as the porosity and WHC of the growing media [45]. Our results show that the biochar input in pots had no significant impact on the initial WHC of the substrate mixture (Table 2), confirming that both peat and biochar had comparable porosity and water retention capacity. The increased WHC by the end of plant growth stage (31 DAS) under 30% biochar treatment (Table 2) may suggest that physical and chemical properties of substrates were changed by fertirrigation.

### 4.2. Plant Physiological Characterization

Germination, as part of an initial growth process, may differ according to plant species and growth media. It may be stimulated by water-soluble organic compounds of the biochar

but retarded by heavy metals or phytotoxic organic compounds. In our study, however, the content of heavy metals in the substrates are below those expected to retard germination. On the other hand, the presence of inhibitory organic compounds and application with the biochar cannot be excluded [46].

A further factor that can be responsible for the observed retardation of the germination can be osmotic stress introduced by elevated pH and EC. Whereas such osmotic stress plays a minor role for the seed development in B15 and B30, it may already affect that in B50 and without doubt that in B70.

However, germination is not only affected by one property of a biochar or the biochar:peat mixture. Thus, one may speculate that for the biochar:peat mixtures B50, the interplay of positive and negative effects puts the pendulum into a negative direction. Nevertheless, a more detailed investigation of the impact of different factors on seed germination is still needed.

The growth parameters analyzedfor lettuce plants showed that, according to biochar: substrate mixtures, the biomass production as well as its leaf area can be increased with up to 30% peat substitution by biochar if compared to the peat substrate. These contents are in agreement with results reported by Awad et al. [47] but are smaller than in other experiments reporting a beneficial impact yield from 40–70%. In our experiments, biochar input above 50% reduces dry shoot biomass (Figure 2B), whereas additional 70% biochar input strongly decreases plant growth and development, as shown by the root to shoot ratio (Figure 2D). In addition, there was no evident stress detected from the chlorophyll content detected by the SPAD parameter for the plants grown on a media with <50% biochar (Table 6). However, first indicators for plant stress were evidenced for the plants grown under the B50 treatment. This tendency describes a bell-shaped curve, indicating that the optimum biochar percentage for plant growth was recorded in the B30 treatment, in comparison to plants grown in the absence of biochar. Above this biochar content, plants start to decrease their growth possibly because of the high alkalinity of the substrate or the imbalance in nutrient availability.

### 4.3. Nutrients Dynamic and Effect in the Substrate-Plant System

In our study, the levels of P, Ca, K, Cu, Zn, and B (Table 3) in the peat:biochar substrate mixtures at the start of lettuce cultivation increased with biochar supply. A similar observation applies for the other nutrients. The fact that the amount of retained nutrient increase with biochar supply implies that biochar plays a major role in this sequestration process [48]. It should be noted that other elements, e.g., B, Na, Cu, or Zn, are potentially toxic and may represent a risk for optimal plant growth, if their concentration surpasses the level which can be tolerated by the respective plant. The EU Regulation 2019/1009 establishes thresholds of 500 and 200 $\mu g\ kg^{-1}$ dry matter for Zn and Cu [49]. With respect to B toxicity, several studies agreed that it may be reached by 30–35 $\mu g\ kg^{-1}$ in substrate, although this would be variable according to plant sensitivity and growing phase [50]. Our results are in agreement with studies showing that Zn and Cu levels above 190 and 150 $\mu g\ kg^{-1}$ are likely to induce a toxic growing environment. A further ion of concern is $Na^+$ since it can cause salinity stress. Nocentini et al. [7] found a negative relationship between $Na^+$ contents in growing substrates amended with biochar and development of tomato plants. High concentrations of Na in the growing media leads to increased uptake by plants. Indeed, we detected a high accumulation of Na in the shoot of 1.48 $mg\ kg^{-1}$ in the plants grown with 30% of biochar, although this level seemed to be still tolerable for lettuce plants. The higher level in the substrate with 50% biochar, however, resulted in decreased plant growth, most likely because it generated osmotic stress conditions, and hindered absorption of essential nutrients such as Ca and K [51]. On the other hand, it is important to consider that high K levels in the substrate can have also suppressive effects on the plant performance and germination [6,52]. In our experiment, although plants grown with more than 30% biochar showed increased K, Cu, Zn, and Na levels when compared to plants grown on peat, no stress symptoms were detected. The accumulation of Ca, Fe,

and B in the shoot decreased as the biochar content increased in substrate, which may also have contributed to the reduced growth and SPAD quantifications. As Atzori et al. [53] have stated that P levels follow similar trends as K levels, our results partly concur with this, showing that P levels maintained higher levels below 50% of biochar addition than K levels.

Concerning the N case in the substrate due to biochar supply is predominately caused by an increase in the content of $NO_3^-$ whereas $NH_4^+$ decreased. Certainly, this also affected the N cycle in the substrate–plant growing system [26,54]. Multiple studies have reported that biochar contribution reduces $NO_3^-$ leaching losses while increasing the nitrification rate after fertigation. However, some of the mechanisms underlying such changes are still scarcely understood [55]. Nevertheless, what can be deduced from our results is the fact that the used biochar increased the immobilization and capture of $NO_3^-$ after continuous fertilizer application, most likely due to its adsorption through suggested mechanisms that involved electrostatic attraction, ion exchange, and surface functional groups [56,57]. To what extent such a N sequestration leads to a competition for N between biochar and plants, however, is still not clear. The utilization of $NO_3^-$ and $NH_4^+$ by plants represents assimilation and immobilization processes, which can be quantified as NUE for plants [58]. Biochar is able to increase plant NUE by taking up $NO_3^-$-N and increasing local $NO_3^-$-N concentration in the plant [59]. This is in line with the increase of NUE values in lettuce plants with increasing biochar addition observed in our study (Table 9). Most likely, the adsorption characteristics of biochar that can be used to enhance the $NO_3^-$-N reduction ability of the root system improve the uptake and utilization of N by the plants. Therefore, possible accumulation of $NO_3^-$ could be attributed to the electrochemical interaction of the basic functional groups of the biochar [60], allowing extra N supplementation when needed. In addition, several studies acknowledge that biochar is able to modify the composition of bacterial communities [61]. This would increase the abundance of functional genes related to N metabolic pathways or N metabolism in the growing media [60,61], which may accelerate both N cycling and availability to crops [62]. However, in our case, this can only be hypothetical. Note that, using 50% biochar input accompanied by $NO_3^-$-N fertilizer still increases NUE compared to unammended plants, but reduces NUE by 1.45-fold if compared to the optimal 30% biochar treatment (Table 9). Therefore, the excess of biochar input indirectly alters N dynamics and physical and chemical properties in the growing media, which ultimately imbalances the environment for growth, nutritional status, and NUE among others [63].

## 5. Conclusions

The substitution of peat by 30% biochar obtained from vineyard prunings (pyrolyzed at a temperature of 500 °C during 4 h, initially open-air full combustion and limited oxygen up to finish) in the plant culture media represents an adequate amount to obtain a growing substrate for cultivating lettuce. More than 50% of biochar in the treatments, on the other hand, is beyond the desirable limit for crop cultivation (depending on the type and pyrolysis temperature of the biochar). This study indicates that the optimal peat replacement by biochar in plant substrates could be achieved with 30% pot volume and gives an understanding of the effect of biochar supply on physical, chemical, and nutritional factors of the substrate–plant system. By testing the recommended biochar percentage on growth and yield quality, we were able to establish an approach allowing higher plant yield and a beneficial supply of nutrients for lettuce plants without detecting plant stress symptoms and improving the performance of crops. The latter may allow a reduction of the addition of selected nutrients and even N to the nutrient solution in hydroponic cultures. Of course, further trials are recommended for future work on a broader range of plants.

**Author Contributions:** Conceptualization, Á.F.G.-R., H.K. and M.A.R.; data curation, Á.F.G.-R., F.J.M.-R. and J.M.G.d.C.B.; formal analysis, Á.F.G.-R., H.K. and M.A.R.; funding acquisition, H.K. and M.A.R.; investigation, Á.F.G.-R.; methodology, Á.F.G.-R., F.J.M.-R., J.M.G.d.C.B., H.K. and M.A.R.; project administration, H.K. and M.A.R.; resources, J.M.C.-F., N.G., H.K. and M.A.R.; supervision,



H.K. and M.A.R.; visualization, M.A.R.; writing—original draft, Á.F.G.-R.; writing—review & editing, Á.F.G.-R., F.J.M.-R., J.M.C.-F., N.G., H.K. and M.A.R. All authors have read and agreed to the published version of the manuscript.

**Funding:** This research was funded by European Union's Horizon 2020 research and innovation programme under the Marie Skłodowska-Curie grant agreement No 895613 and EIT Food program (Black to the Future Project, EIT-21217). This EIT Food activity has received funding from the European Institute of Innovation and Technology (EIT), a body of the European Union, under Horizon Europe, the EU Framework Programme for Research and Innovation. Á.F. García-Rodríguez acknowledges the Spanish National Research Council for providing JAE Intro-ICU grant.

**Data Availability Statement:** The data presented in this study are available on request from the first author and the corresponding author.

**Acknowledgments:** Acknowledgements go also to Gruppo Caviro for the purchasing and shipping of the applied biochar. M. Velasco is gratefully acknowledged for her technical assistance. We acknowledge support of the publication fee by the CSIC Open Access Publication Support Initiative through its Unit of Information Resources for Research (URICI).

**Conflicts of Interest:** The authors declare no conflict of interest. The funders had no role in the design of the study; in the collection, analyses, or interpretation of data; in the writing of the manuscript; or in the decision to publish the results.

**Disclaimer:** This manuscript does not reflect the views of the European Union.

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
