# Peer review of "Influence of Biochar Mixed into Peat Substrate on Lettuce Growth and Nutrient Supply"

_horticulturae, doi:10.3390/horticulturae8121214_

Round 1
Reviewer 1 Report
Journal: Horticulturae
Title: Partial substitution of peat by biochar in plant growing substrates increases nutrient supply and promotes growth of lettuce plants
MS No. horticulturae-2014746
The authors replaced peat by biochar obtained from vineyard pruning in plant growing substrates, and studied its effect on growth of lettuce plants under greenhouse semi-hydroponic conditions. The results demonstrates that the used biochar can reduce the need of specific nutrients in the nutrient solution of horticultural cultures and replace peat in growing media by up to 30% with a positive impact on plant growth.
The topic of this paper is interesting and the results are useful in practice. At large, the experiments were well designed and the paper was prepared well. However, it seems that the mechanisms of the results were somewhat unclear. The paper could be accepted after a minor revision.
1. Too lengthy for INTRODUCTION;
2. The mechanisms are not clear;
3. P3, only nitric acid was used to digest the sample?
4. On what basis, the ratios were arranged in Section 2.3?
5. P6, too lengthy for tables’ title. Maybe, some of them should be described in experimental section or table note should be used;
6. The Refs are so many. It is nor proper.
Author Response
We appreciate the comments and recommendations of the both editors regarding our paper. We are grateful that the topic of this paper fits perfectly in this journal. We considered all suggestions and hope that with the requested corrections the paper is in the shape for being accepted for publications. As recommended we asked a native English speaker with experiences in English editing for revision and included his suggestion. Hereafter, we detail the aspects that have been corrected after a carefully review:
Reviewer 1:
- Too lengthy for INTRODUCTION;
Answer: We have taken into consideration your advices concerning the length and writing of this paper, and so we have eliminated the unnecessary and repeated parts within introduction, results, discussion and bibliography. Although, the paper conceals now the essential information in order to fully understand all the aspects analyzed and discussed.
- The mechanisms are not clear;
Answer: We are sorry, but it is not clear to us which mechanisms were questioned by the reviewer, but we revised those which seemed for us to need a better explication.
- P3, only nitric acid was used to digest the sample?
Answer: To digest the sample in order to analyze sample by ICP-OES ultrapure nitric acid for plant samples and for substrates sample hydrochloric and nitric was used to controlled acidic digestion as described in the text.
- On what basis, the ratios were arranged in Section 2.
Answer: The ratios were arranged according to pot volume. Thus, the increase of the volume of the biochar content went along with decrease of the peat content by a constant volume of vermiculite. We improved the description in section 2.3
- P6 too lengthy for tables’ title. Maybe, some of them should be described in experimental section or table note should be used;
Answer: We appreciate the comment and indeed the heading were long. Being aware that table titles should contain all information necessary to understand the table without the manuscript text, we decided to put the required info into the table notes.
- The Refs are so many. It is nor proper.
Answer: We have shortened the reference list to a total of 64 in order to fully explain and support our results.
Reviewer 2 Report
The topic offered by the Authors fits perfectly within the journal's fields of interest. The proposed experimental trial was aimed at replacing the use of peat with biochar as a plant-growing substrate, being the former a non-renewable resource, rich in sequestered Carbon that should be preserved and not consumed, in the perspective of climate change mitigation.
The experimental set was planned and carried out by adopting a complete and in-depth array of analyses, considering both the physical and chemical parameters of the compared substrates (peat vs biochar), as well as plant indicators relating to their growth and physiological status.
It was planned to test different proportions of biochar in the mixed substrate composition, therefore it was possible to effectively check the optimal amount of biochar to add to the mix, based on the complete set of variables under observation, thus obtaining the best results in terms of plant growth, quality of plants and nutrient use efficiency.
For these reasons, the submitted manuscript should be considered worthy of being published. Unfortunately, several flaws prevent this happy conclusion, so far.
Detailed comments, suggestions, and advice are reported in the attached copy of the manuscript where I have added several notes along the text (to be considered very carefully).
The following are general but fundamental comments in order to work out a new and improved version of the manuscript that, considering my point of view, deserve a deep and thoughtful improvement.
1. A wide range of results is worthwhile, but it is equally important to be able to represent them without lengthening the reading time too much. You should take care of the easiness of being read and understood.. Too many details, sometimes obvious, are harmful and add nothing in credit to the Authors.
2. Please, avoid overlapping and repetitions between the two consecutive sections, "results" and "discussion", respectively. Repetitions weigh down the reading and are tiring and boring.
3. It is necessary to improve significantly the quality of the English applied in the text. Some sentences are difficult to understand, also (but not only) because of the poverty of expression in the English language. Be clear, concise, and straightforward, adopting well-structured and short sentences.
4. Please, when considering “vineyard pruning” we are not in the sector of “waste”, we are still in the sector of “agricultural residues”, therefore we have to refer to this kind of feedstock as a “byproduct”. This is also in line with EU regulations.
5. It could be very useful a characterization of the used biochar according to currently available European and international standards (such as EBC “European Biochar Certificate” or IBI Biochar Standards).
6. Your manuscript does not show on the left side of the page the line numbering thus making troubles for an easy revision. Do not forget this practical rule next time.
7. In the section dedicated to the applied statistical analysis you report to have performed both a “one-way analysis of variance” (ANOVA) and a multivariate analysis of variance (MANOVA). To be honest, there is no evidence that you performed a MANOVA. What you have called a MANOVA is, instead, a “two-way” ANOVA (Treatments and Harvest Time, together with their interaction).
8. Considering the figures reported in all the tables, a systematic error should be emphasized. The average value and its associated uncertainty (experimental error) must always have the same number of digits after the decimal point. If the uncertainty has more places after the decimal as compared to the reported average, adding it to (or subtracting it from) the average will leave the resulting number with more decimal places than your original measure. For example, 2.4 ± 0.16 implies that the result lies in the range 2.24 – 2.56. But the reported average has a precision only up to one place after the decimal. Hence the correct way to express the average is 2.4 ± 0.2.
If the uncertainty has less number of places after the decimal than the average, then your best estimate is rounded off appropriately. This is because even if the reported average is very precise, it does not have that accuracy (as proved by the error having less number of places after the decimal). Thus, 2.456 ± 0.12 should be written as 2.46 ± 0.12. Please, check all your tables and report the average according to this instruction.
9. When you construct an ANOVA table, the mean separation has been performed considering a “pooled” error term (the residual variance of the ANOVA). Therefore, there is no need to report the standard error (±SE) of each mean in the table. Conversely, it is possible, at the bottom of each column of means, to report the pooled SE or, alternatively, the HSD (“Honest Significant Difference”) or the LSD (“Least Significant Difference”) or whatever you apply in performing the mean separation (Duncan’s test, etc.).
10. The table headings are long and confusing. They should be shortened and made more readable. It is not necessary to report all the details when they are so evident.

Author Response
We appreciate the comments and recommendations of the both editors regarding our paper. We are grateful that the topic of this paper fits perfectly in this journal. We considered all suggestions and hope that with the requested corrections the paper is in the shape for being accepted for publications. As recommended we asked a native English speaker with experiences in English editing for revision and included his suggestion. Hereafter, we detail the aspects that have been corrected after a carefully review:
Reviewer 2:
The topic offered by the Authors fits perfectly within the journal's fields of interest. The proposed experimental trial was aimed at replacing the use of peat with biochar as a plant-growing substrate, being the former a non-renewable resource, rich in sequestered Carbon that should be preserved and not consumed, in the perspective of climate change mitigation.
The experimental set was planned and carried out by adopting a complete and in-depth array of analyses, considering both the physical and chemical parameters of the compared substrates (peat vs biochar), as well as plant indicators relating to their growth and physiological status.
It was planned to test different proportions of biochar in the mixed substrate composition, therefore it was possible to effectively check the optimal amount of biochar to add to the mix, based on the complete set of variables under observation, thus obtaining the best results in terms of plant growth, quality of plants and nutrient use efficiency.
For these reasons, the submitted manuscript should be considered worthy of being published. Unfortunately, several flaws prevent this happy conclusion, so far.
Detailed comments, suggestions, and advice are reported in the attached copy of the manuscript where I have added several notes along the text (to be considered very carefully).
The following are general but fundamental comments in order to work out a new and improved version of the manuscript that, considering my point of view, deserve a deep and thoughtful improvement.
- A wide range of results is worthwhile, but it is equally important to be able to represent them without lengthening the reading time too much. You should take care of the easiness of being read and understood. Too many details, sometimes obvious, are harmful and add nothing in credit to the Authors.
Answer: We have deeply revised, modified and substituted recommended words from the revised paper and thereby included them into the new version. We have also taken into consideration your advices concerned the length and writing of this paper, and have eliminated the unnecessary and repeated parts within introduction, results, discussion and bibliography. In addition, we involved a native English speaker with experience in manuscript editing and included his suggestions to improve the writing. We think that we achieved that now the paper conceals the essential information.
- Please, avoid overlapping and repetitions between the two consecutive sections, "results" and "discussion", respectively. Repetitions weigh down the reading and are tiring and boring
Answer: We have checked Results and Discussion and we did our best to remove overlappings It is necessary to improve significantly the quality of the English applied in the text. Some sentences are difficult to understand, also (but not only) because of the poverty of expression in the English language. Be clear, concise, and straightforward, adopting well-structured and short sentences.
- It is necessary to improve significantly the quality of the English applied in the text. Some sentences are difficult to understand, also (but not only) because of the poverty of expression in the English language. Be clear, concise, and straightforward, adopting well-structured and short sentences.
Answer: We appreciate your comments regarding the writing and recommendations to be direct, easy to be read understood. In addition, an English native college has carefully checked the paper.
- Please, when considering “vineyard pruning” we are not in the sector of “waste”, we are still in the sector of “agricultural residues”, therefore we have to refer to this kind of feedstock as a “byproduct”. This is also in line with EU regulations.
Answer: We have eliminated the term “waste” regarding vineyard pruning’s as we understand that in the sector of “agricultural residues” has to be referred to this kind of feedstock as a “byproduct”
- It could be very useful a characterization of the used biochar according to currently available European and international standards (such as EBC “European Biochar Certificate” or IBI Biochar Standards).
Answer: We obtained this biochar from a company which used this biochar for an EU project and did the biochar characterization. We will forward your recommendation to the company.
- Your manuscript does not show on the left side of the page the line numbering thus making troubles for an easy revision. Do not forget this practical rule next time.
Answer: We used the template of the journal and somehow the numbers disappeared. They are now included.
- In the section dedicated to the applied statistical analysis you report to have performed both a “one-way analysis of variance” (ANOVA) and a multivariate analysis of variance (MANOVA). To be honest, there is no evidence that you performed a MANOVA. What you have called a MANOVA is, instead, a “two-way” ANOVA (Treatments and Harvest Time, together with their interaction).
Answer: We agree that the performed statistics were “One-Way” and “Two-Way” ANOVA, to check interaction between Treatments (T) and Harvest Time (HT) variables. Thus we eliminated the word MANOVA from the text to avoid confusion.
- Considering the figures reported in all the tables, a systematic error should be emphasized. The average value and its associated uncertainty (experimental error) must always have the same number of digits after the decimal point. If the uncertainty has more places after the decimal as compared to the reported average, adding it to (or subtracting it from) the average will leave the resulting number with more decimal places than your original measure. For example, 2.4 ± 0.16 implies that the result lies in the range 2.24 – 2.56. But the reported average has a precision only up to one place after the decimal. Hence the correct way to express the average is 2.4 ± 0.2.
If the uncertainty has less number of places after the decimal than the average, then your best estimate is rounded off appropriately. This is because even if the reported average is very precise, it does not have that accuracy (as proved by the error having less number of places after the decimal). Thus, 2.456 ± 0.12 should be written as 2.46 ± 0.12. Please, check all your tables and report the average according to this instruction.
Answer: Thanks for the comment: We have precisely checked error in decimal criteria for each table
- When you construct an ANOVA table, the mean separation has been performed considering a “pooled” error term (the residual variance of the ANOVA). Therefore, there is no need to report the standard error (SE) of each mean in the table. Conversely, it is possible, at the bottom of each column of means, to report the pooled SE or, alternatively, the HSD (“Honest Significant Difference”) or the LSD (“Least Significant Difference”) or whatever you apply in performing the mean separation (Duncan’s test, etc.).
Answer: We considered your advices but prefer to maintain the SE for being more clear. The table headings are long and confusing. They should be shortened and made more readable. It is not necessary to report all the details when they are so evident.
- The table headings are long and confusing. They should be shortened and made more readable. It is not necessary to report all the details when they are so evident.
Answer: We shortened the titles as described above for the comments of reviewer 1.
Comments within the paper:
We considered the suggested changes marked in the paper. All word changes are now included into the new version. In addition, we changed some wording or typos were we found it necessary.